# Cryopreservation of *Malus* and *Pyrus* Wild Species in the ‘Fruit Genebank’ in Dresden-Pillnitz, Germany

**DOI:** 10.3390/biology12020200

**Published:** 2023-01-28

**Authors:** Monika Höfer, Henryk Flachowsky

**Affiliations:** Julius Kühn-Institute (JKI), Federal Research Centre for Cultivated Plants, Institute for Breeding Research on Fruit Crops, Pillnitzer Platz 3a, 01326 Dresden, Germany

**Keywords:** dormant buds, genetic resources, liquid nitrogen, long-term storage, pome fruit, vitrification

## Abstract

**Simple Summary:**

In Dresden-Pillnitz, Germany, the genebank preserves samples of wild apple and pear species in addition to fruit varieties in field collections. To protect this unique and valuable diversity from pests, diseases, and weather extremes, collections of duplicates in a different location are necessary. Cryopreservation, storage at ultralow temperature (usually in liquid nitrogen at –196 °C), is used for this purpose. Two different methods were applied; direct dormant bud cryopreservation and cryopreservation of shoot tips from in vitro culture. In apple, a total of 180 samples belonging to 32 species with an average survival rate after cryopreservation of 39% within ten years were processed using dormant buds. Samples numbering 116 reached the 40% survival criterion, which corresponds to successful treatment. Therefore, the dormant bud technique will become the method of choice for wild apple species in the future. In the case of pear wild species, a total of 35 samples belonging to 21 species using both techniques for cryopreservation were tested. In these experiments, the method using shoot tips showed better results compared to the dormant buds. Further trials with both techniques are planned for the future in order to make a final decision for the preservation of pear wild species.

**Abstract:**

A unique and valuable diversity of the *Malus* and *Pyrus* wild species germplasm is maintained ex situ in field collections in the ‘Fruit Genebank’ in Dresden-Pillnitz, Germany. (1) Background: The establishment of a duplicate collection is necessary to preserve this material safely from abiotic and biotic stress factors. (2) Methods: Two different techniques, cryopreservation using dormant buds and PVS2 vitrification using in vitro shoot tips, were applied and compared. (3) Results: In *Malus* altogether 180 accessions belonging to 32 species were processed with an average recovery rate of 39% within ten years using the dormant bud method. Accessions, 116 in number, achieved the criterion of 40% recovery which was 64.44% of all accessions tested. In the case of *Pyrus* germplasm a total of 35 accessions of 21 species and both techniques for cryopreservation were tested. In the results of these experiments the PVS2 method led to better results compared to the dormant bud method. (4) Conclusions: In *Malus* the dormant bud technique will be the method of choice for the future to build up the duplicate collection. Further experiments using both techniques are planned in the future to make a final decision for *Pyrus*.

## 1. Introduction

Conservation of fruit genetic resources is a pre-requisite for breeding new varieties for sustainable fruit production [1]. For this reason, the first government-organized activities were started in Germany very early on in the early decades of the 20th century. The beginning of scientifically based fruit breeding in the early 1920s and the establishment of the first institutes for breeding research made it necessary to build up extensive collections of plant genetic resources. The first institutional collections were established in Müncheberg and Naumburg (both in Germany), which mainly consisted of landraces, primitive forms, and wild species. These collections were used as the basis for science-based fruit breeding in the following years [2]. The collections often consisted of plants that had been collected decades earlier through exchanges with other Arboreta. In many cases, accurate information on the origin was missing and the original passport data was incomplete or not available. Information on the country of origin, the original collector, and the habitat in which the genetic resource was taken from were often lacking.

After interruption during World War II, all fruit breeding activities in the territory of the former German Democratic Republic (GDR) were centralized at the Institute for Fruit Research in Dresden-Pillnitz. This also entailed a relocation of the collections from Müncheberg and Naumburg. Extensive plant material of both collections was transferred to Dresden-Pillnitz and preserved in the still existing ’Fruit Genebank’. The ex situ field collections of cultivated and wild species of several genera (e.g., *Malus*, *Pyrus*, *Prunus,* and *Fragaria*) have since been maintained at Pillnitz [2].

Since 2003, the ‘Fruit Genebank’ has belonged to the Federal Research and is maintained at the Institute for Breeding Research on Fruit Crops, which is one of the eighteen research institutes of the Julius Kühn-Institut (JKI). The ‘Fruit Genebank’ is not only focusing its activities on conservation but has started to systemically restructure the collections and evaluate the existing plant material. Today, the ‘Fruit Genebank’ contains approximately 4500 accessions of cultivated fruit species and their wild relatives.

Most of the temperate fruit species are cross-pollinated. Their genome is heterozygous and they are propagated vegetatively. Fruit genetic resources are either maintained in the field as active collections or as passive collections in vitro or in cryopreservation.

In the ‘Fruit Genebank’ in Dresden-Pillnitz preservation of the existing unique and valuable diversity of fruit germplasm is realized in an ex situ field collection on about 10 ha of land. Here, each cultivar and each wild species accession is kept in two trees. However, several drawbacks limit the efficiency and jeopardize the security of these collections. For example, annual infections with pathogens like the fire blight bacterium *Erwinia amylovora* cause damage to apple and pear orchards and kill susceptible genotypes [3,4]. Backups for these materials are urgently needed to provide security. The cultivars of the ‘Fruit Genebank’ are duplicated within the framework of the German Fruit Genebank, a decentralized conservation network [5]. According to the conservation strategy of the German Fruit Genebank for all fruit species, each cultivar should be maintained in at least one duplicate for safety in the established fruit specific networks.

Since wild species accessions are not part of the network of the German Fruit Genebank, another way to secure them has to be found. Duplication of the entire collection at a second location in Germany is not a realistic solution due to the high operating costs and budget limitations. On the other hand, seed collections do not represent the unique genotypes of clonal heterozygous accessions. In vitro gene banks offer an alternative form for the conservation of genetic resources in a number of crops [6]. Despite numerous advantages, in vitro gene banks are also not ideal for long-term storage of a large collection because, plantlets require repropagation at intervals and may be lost due to contamination or technical difficulties [7].

For the preservation of such genebank collections, cryopreservation is the method that can be used to circumvent many of the disadvantages mentioned above. However, wider application of plant cryopreservation depends on the availability of efficient and reproducible protocols applicable to many different plant species [8]. The development of cryopreservation techniques for clonal woody species began in the late 1970s by the work of Sakai and Nishiyama [9]. Further, a range of cryopreservation techniques was developed and used for different genera and species [10] and is now a useful method for the long-term storage of germplasm. Benelli et al. [11] reviewed the advances made over the last decade in cryopreservation of economically important, vegetatively propagated fruit trees. Cryopreservation protocols have been established using both dormant buds sampled from field-grown plants and shoot tips sampled from in vitro plantlets [12].

The clonal wild-species *Malus* collection of the Julius Kühn-Institute, Institute for Breeding Research on Fruit Crops, Germany, comprises 578 accessions of 49 species and represents one of the largest collections in Europe. Two hundred and fifteen accessions were obtained from the historical collection of the former Institute for Phytopathology of the Biological Central Centre at Naumburg, Germany. The Nikolaj I. Vavilov Research Institute of Plant Industry, St. Petersburg, Russia and the USDA-ARS Plant Genetic Resources Unit, Cornell University, Geneva, USA also provided accessions. Furthermore, the *Malus* material was expanded substantially through expeditions into the centers of origin [13,14,15]. The *Pyrus* collection comprises 88 accessions of 29 species. It is also being expanded by field expeditions, especially for the species *Pyrus pyraster*, which is native to Germany.

The present work describes the development and implementation of cryopreservation as part of the conservation strategy for germplasm of *Malus* and *Pyrus* spp. wild species under the conditions available in Dresden, Germany. Two different techniques, cryopreservation by using dormant buds and PVS2 vitrification by using in vitro shoot tips were compared.

## 2. Materials and Methods

### 2.1. Material

Plant material was collected in the ‘Fruit Genebank’ at the Institute for Breeding Research on Fruit Crops, Dresden-Pillnitz. The institute is located at 51°00′07″ N latitude, and 13°52′59″ E longitude, altitude 115 m, with 9.1 °C annual average temperature and 668 mm annual precipitation. The soil type at the orchard is clayey sand with pH 5.6–6.6.

Scion wood containing the current season’s growth was cut from the clonal *Malus* and *Pyrus* wild species collection. Screening and storage of dormant buds was conducted over a period of 10 years for 180 *Malus* wild species accessions. Material from the apple cultivar ‘Idared’ was used as a control variety in all years to test the complete process. In addition, the preliminary results for 33 *Pyrus* accessions are also presented here. The genotypes cryopreserved each year were randomly selected. Wood was collected in January when the temperature was between 0 and −5 °C for at least 72 h. Afterwards, the branches were stored in plastic bags at −5 °C ± 1 °C for a minimum of six days.

The *Pyrus* donor material for the PVS2-experiments was well-established in vitro cultures of 14 accessions. For culture initiation 3-cm cuttings were collected from trees in the orchard at the beginning of spring. Two media composition were used both for initiation and proliferation in dependence of the genotype: (1) Murashige and Skoog (MS) [16] medium with 0.44 µM benzyl amino purine (BAP), 0.054 µM 1-naphthaleneacetic acid, 0.29 µM gibberellic acid, 30 g/L sucrose, and 0.75% agar (2) A modified MS medium according to Cheng [17] with 4.4 µM BAP, 30 g/l sucrose, 0.35% agar, and 0.18% gelrite in 200 mL glass jars (40 mL/jar). Growth room conditions were 23 °C with a 16-h photoperiod under 60–65 μmol m s^−1^ photon flux. Apical shoot tips were excised from three to four-week-old in vitro plants for cryopreservation.

### 2.2. Cryopreservation—Dormant Buds of Malus and Pyrus

For *Malus* spp., a protocol for cryopreservation of dormant apple buds was modified by Höfer [18] from one developed at Fort Collins, Colorado [3]. The bud wood was wrapped in plastic bags and stored at −5 °C for at least 6 days until further processing. Stem sections 35 mm long, with only one bud, in the middle, were cut from dormant scions. These sections were dehydrated to 30% moisture on large-mesh, metallic trays in a −5 °C cold chamber. The percent moisture was determined by gravimetric measurement. Two or three sections were then placed in 4.5-mL cryovials Nunc™, Thermo Fisher Scientific^®^, Roskilde, Denmark) in a controlled-rate temperature reduction freezer (Kryo 360-3.3, Messer Cryotherm, Euteneuen, Germany) using the freezing protocol of 1 °C/h to −30 °C. After holding at −30 °C for 24 h, the cryovials were transferred into cryoboxes and stored for 2 months in the vapor phase over liquid nitrogen (LN; (Biosafe, Messer Cryotherm, Euteneuen, Germany). For recovery, vials were rewarmed to +4 °C in a refrigerator for 24 h. For rehydration, the sections from each vial were placed separately in trays of wet, autoclaved sand at 4 °C for 15 days. The chip budding technique (double-budded) was used to graft the sections onto 1-year-old M9 apple rootstocks planted in the orchard. Recovery data were taken 5 months after grafting.

For *Pyrus* the same method was used. The rootstock used was the Kirchensaller Mostbirne.

### 2.3. Cryopreservation—PVS2 Vitrification of Pyrus

For all steps of cryopreservation with PVS2, the MS-media described in Section 2.1 were used as the basic medium, depending on the genotype. After the last subculture, 2- week-old shoots of *Pyrus* were cold acclimated for 14 days (16 h at −1 °C darkness and 8 h at 22 °C light) [19]. After cold acclimation, shoot tips were then dissected and cultured on MS medium (Duchefa, Haarlem, The Netherlands) supplemented with 5 % (*v*/*v*) dimethyl sulfoxide (DMSO; neoLab, Heidelberg, Germany) and 1 g of additional agar (0.85 % [*w*/*v*]; DifcoTM Agar granulated, Becton Dickinson and Company, Sparks, NV, USA) for 2 days under cold-acclimation conditions. Subsequently, the shoot tips were incubated in loading solution (2 M glycerol [neoLab, Heidelberg, Germany] and 0.5 M sucrose [Grade II, Sigma-Aldrich^®^, München, Germany] in MS medium) for 15 min at room temperature 23 °C and finally transferred to 1.8-mL cryovials (Nunc™, Thermo Fisher Scientific^®^, Roskilde, Denmark). The explants were incubated on ice for 2.5 h [20] with 0.75 mL Plant Vitrification Solution 2 (PVS2: 30 % [*w*/*v*] glycerol neoLab, 15% [*w*/*v*] ethylene glycol (neoLab), 15% [*w*/*v*] DMSO (neoLab), 0.4 M sucrose [Sucrose Grade II, Sigma-Aldrich^®^]) in MS medium [21]. Finally, the cryovials were plunged directly into LN in a cryotank (Biosafe, Messer Cryotherm, Kirchen, Germany) for 1–3 days. Each vial was filled with five shoot tips.

In order to test the recovery of the plant material stored in LN, the vials were rewarmed by plunging them into sterile water (40 °C) for 2 min after 1–3 days of storage. After rewarming, the PVS2 was partly drained and replaced twice by 1.2 M sucrose (Sucrose Grade II, Sigma-Aldrich^®^, München, Germany) at 25 °C. Subsequently the shoot tips were transferred to Petri dishes (Ø 55 cm, Carl Roth, Karlsruhe, Germany) containing the same proliferation medium. Shoot tips in Petri dishes on proliferation medium were moved to the growth room for 1 week in darkness, and then moved immediately into the light conditions described above. Assessments of the recovery of the shoot tips were conducted at different times after rewarming, beginning after two weeks. Shoot survival, proliferation, leaf expansion, and greening were considered in assessing successful recovery from cryopreservation

### 2.4. Statistical Analysis

For the dormant bud experiments, 50 single-node sections were regularly used as a representative sample per genotype; of these, 30 sections were held for long-term storage and 20 sections were processed for recovery tests. Each vial was filled with two or three dormant buds, depending on the diameter of the buds; therefore, the long-term samples of a genotype consisted of 10–15 vials. For *Malus,* each specific accession was tested for a second year when regeneration was 0%. For *Pyrus*, all experiments so far have only been carried out once due to the technical requirements.

Each PVS2 vitrification experiment included 40 cryopreserved shoot tips and 20 sections which were processed for recovery tests. For generation of control variants, (same procedure without storage in LN) 10 shoot tips were used in each case. In the first year of the experiment, the influence of the size of the explant on the regeneration rate after cryopreservation was tested with eight genotypes in a separate experiment. The variants (each 20 shoot tips) were classified according to the number of primordia; explants with meristem and one primordium, with two, with three, or with four primordia.

The mean recovery rate and its standard deviation were calculated across the accessions of each species (Microsoft Office Excel 2007). ANOVA and Duncan’s Multiple Range Test (*p* ≤ 0.05) using SAS Enterprise Guide 4.3. were applied to data analysis.

## 3. Results

### 3.1. Cryopreservation of Malus—Dormant Bud Experiments

To apply the optimized protocol for cryopreservation of *Malus* genetic resources [18] to further accessions of the *Malus* field collection of the JKI’s ‘Fruit Genebank’ a total of 180 accessions of different *Malus* wild species was cryopreserved over a period of 10 years from 2011 to 2021 (Table 1). A minimum recovery of 40% was determined as the baseline for storage of a given accession [22].

Altogether 180 *Malus* accessions belonging to 32 species were processed. For *Malus* spp., the average recovery rate of long-term cryopreserved dormant buds was 39%. Of the 180 accessions cryopreserved, 116 (64.44%) had viability levels greater than 40%. Due to laboratory conditions, the number of samples per year was limited to about 30 accessions. These accessions were randomly selected. The average time of drying to a water content of 30% over all genotypes per year varied between 4 and 15 days (data not shown).

As a control, the experiment with the apple cultivar ‘Idared’ was carried out every year. The initial moisture content varied from 50.61 to 35.01%, so the time needed to achieve 30% moisture content ranged from 22 to 3 days. The recovery rates for the cultivar ‘Idared’ are on average 86.35 ± 9.4% over the experimental years (data not shown).

### 3.2. Cryopreservation of Pyrus—Dormant Bud Experiments

Altogether 33 *Pyrus* accessions belonging to 19 species were processed in the years 2020–2022 (Table 2 and Figure 1). The average recovery rate of cryopreserved dormant buds was 25.45 %. Again a minimum recovery of 40% was determined as the baseline for storage of a given accession [22]. Of the 33 accessions cryopreserved, 12 had viability levels greater than 40%, which is 36.36% of the accessions.

The initial moisture content varied from 56.24 to 33.78% in the three experimental years. So, the time needed to reach 30% content ranged from 41 to 3 days (data not shown).

### 3.3. Cryopreservation of Pyrus–PVS2 Vitrification

First PVS2 experiments were performed in parallel to the dormant bud experiments (Figure 2). In addition to testing different genotypes, the influence of explant size was also investigated during the first year.

After cryostorage and transfer of the shoot tips to the regeneration medium, the first regenerates can be observed after 2 weeks. In the following 2 weeks, about 74% of the possible regenerates are already visible. The maximum number of regenerates is reached after 8 to 10 weeks (data not shown).

In the first year of the experiment, the influence of the size of the explant on the regeneration rate after cryopreservation was tested with eight genotypes in a separate experiment (Figure 3). The average of the tested genotypes demonstrates that the explants with one or two primordia show a significantly higher regeneration rate. Usually, pear explants are approximately 1 to 1.5 mm in size. For this reason, only explants where the meristem was covered by one to two primordia were used in subsequent experiments.

In further experiments, additional genotypes were included and, if sufficient experimental material was available, controls were carried out without cryostorage. Altogether 14 *Pyrus* accessions belonging to 12 species were processed in the years 2019–2022 (Table 3). The average recovery rate of long-term cryopreserved shoot tips was 32.67%. When a minimum recovery of 40% is used as the baseline for storage of a given accession [22], 6 of the 14 cryopreserved accessions, had viability levels greater than 40%. This corresponds to 42.86% of all tested accessions.

Table 4 shows the summary comparison of the experiments carried out in *Pyrus* depending on the methods used, taking into account only the 12 accessions used in both techniques.

For the seven genotypes tested, PVS2 vitrification is the more effective method, while for the two genotypes the dormant bud method shows the greater percentage values of regeneration. For another three genotypes, no conclusion could be drawn regarding the more effective method.

## 4. Discussion

The aim of the study was to summarize the attempts to develop an effective conservation strategy for the genetic resources of *Malus* and *Pyrus* wild species in Germany. The philosophy of the work is to develop a system suitable for a wide range of genotypes in the genebank; not for individual genotypes. The development of an effective method for cryopreservation is required for cost-effective long-term storage.

### 4.1. Cryopreservation of Malus

For *Malus* spp., the original protocol for cryopreservation of dormant buds developed at Fort Collins, Colorado [3] was modified for the mild-winter conditions of Central Europe and for the existing laboratory conditions by Höfer [25]. The most important modifications were (i) the additional cold storage of the bud wood in plastic bags at −5 °C before handling, (ii) the freezing and storage in cryotubes, (iii) rehydration in wet, autoclaved sand and (iv) the grafting onto rootstocks in the experimental field. Subsequently, a wide range of accessions representing species belonging to all sections of the genus *Malus* were tested within the genotype screening experiments. In these studies, a relationship between the viability after cryopreservation and the affiliation to a taxonomic group was discussed [18].

Based on the studies presented in this article a total of 180 *Malus* accessions belonging to 32 species were processed with an average recovery rate of 39% within ten years (Table 1). Out of these 180 accessions, 116 had viability levels greater than 40%, which corresponds to 64.44% of all accessions tested. The relationship between viability after cryopreservation and taxonomic groups could be confirmed, even if a large variability was found in the Malus sections. Comparing our results of genotype screening (Table 1) with experiments carried out in the USDA-ARS National Center for Genetic Resources Preservation in Fort Collins, the percentage of tested accessions that met the criterion of 40% recovery was much higher in the U.S. [26,27]. However, our experiments confirmed very good regeneration rates of the species *M. pumila*, *M. baccata*, *M. prunifolia,* and the species of the Section Chloromeles. The species classified by Towill et al. [24] with moderate sensitivity, such as *M. yuanannesis*, *M. sargentii,* or *M. halliana* showed no or very low regeneration rates in the current experiments (Table 1). The species recorded by Jenderek et al. [27] with the highest percentage of accessions with a regeneration of 0% are in agreement with our results for these species. The regeneration rate of zero for the accessions of *M. tschonoskii* is confirmed in all trials.

Despite the fact that the dormant buds present an experimental system with uninfluenceable sources of variation, it is a method that can be used for a wider range of *Malu*s genotypes and is less labour intensive. As there are often annual weather fluctuations, at least two years of testing are suggested for *Malus* accessions. Currently, 173 (app. 30%) wild species accessions are in the long-term storage of cryopreservation in the ‘Fruit Genebank’. Routinely, 50 bud sections per accession are processed, 30 for permanent conservation and 20 for recovery testing. The storage records link the cryopreserved samples to all information about the original plant (passport information) and to the processing data, the number of buds processed, and the percentage viability (number of viable buds determined by grafting). Furthermore, the organization of the Dewar vacuum flasks, where the collection is stored, is maintained in a logbook [28].

For accessions with recovery rates below 40%, other possibilities should be considered for future storage [17]. When low recovery cannot be overcome, more buds of each accession have to be stored. This is to ensure that enough buds will be viable after thawing and to guarantee a safe preservation and successful re-establishment of the genotype [29]. Another possibility would be to regenerate the cryopreserved buds in in vitro culture or, in general, to carry out cryopreservation of shoot tips for specific genotypes. Furthermore, the ‘Fruit Genebank’ is also interested in maintaining specific accessions ex situ in other gene banks and botanic gardens under cooperative agreements. The possibilities of cryopreservation, which include a phase of in vitro culture, should only be considered if all other described ways of duplicate build-up are not possible for selected Malus accessions. The method of cryopreservation using dormant buds will always be the method of choice due to the size of our Malus collection and the comparatively lower labour input.

### 4.2. Cryopreservation of Pyrus

In analogy to *Malus*, the dormant bud protocols developed by Forsline [3] and co-workers, modified by Höfer [18,25] was used in the first trials for cryopreservation of *Pyrus*. Based on the preliminary results present in this study, a total of 33 *Pyrus* accessions belonging to 19 species were processed with an average recovery rate of 25.45% within two years (Table 2). The relationship of viability after cryopreservation to taxonomic groups could also be demonstrated. The literature data on the cryopreservation of dormant buds of *P. communis* cultivars also show strong variations between 0 and 90% [30] and between 0 and 83% [31], respectively. Literature data of various *Pyrus* wild spp. are only available from the USDA-ARS National Center for Genetic Resources Preservation, Fort Collins [27]. Forty accessions (25 taxa) were tested, with 57.55% achieving the criterion of 40% recovery. Our data shows a lower percentage obtained in the first trials (Table 2). Of the 33 accessions cryopreserved, 12 had viability levels greater than 40%, which corresponds to 36.36% of all accessions tested. The effect of the growing season is evident in the experiments of Bilavcik et al. [31]. For this reason, the trials should be repeated in a second year, as with *Malus*. Guyader et al. [32] found strong variation after cryopreservation for the cultivar ‘Williams’. The authors identified several factors, which could significantly affect the success. Among them were factors like bud morphology, rehydration phase (technique used and duration), rootstock genotype, type of grafting, etc. In addition, tests on grafting ability could also provide useful information, which, apart from quality of the rootstocks, also influences the outcomes of regrowth [33].

In *Pyrus*, there was an opportunity to test another method for cryopreservation, as some accessions were already present in in vitro culture. Based on the successful work on cryopreservation of *Fragaria* [20] in our lab., the first work on PVS2 vitrification in *Pyrus* was realized. A total of 14 *Pyrus* accessions belonging to 12 species were processed with an average recovery rate of 32.67% (Table 3), 42.86% of the accessions achieving the criterion of 40% recovery. Comparative experiments with accessions of different *Pyrus* species also showed strongly varying regeneration rates between 5 to 95% depending on the genotype [34]. However, the survival rates could not be linked to the geographical origin of the species tested.

There are several key factors to promote regrowth following cryopreservation. High survival rates of in vitro grown material are expected not only by the cryogenic protocol itself, but also by the physiological conditions of the material to be cryopreserved, such as growth stage, size, and preculture conditions. The age of the stock cultures significantly affects the shoot recovery rate from cryopreserved shoot tips [35]. The age of the plants was not changed in the present trials, but great care was taken to ensure that only in vitro cultures with very good vigor were used as donor material. This observation is also confirmed by Reed, lack of vigor may be contributed to lower tolerance of dehydration, cooling, and regrowth after thawing [34].

Shoot tip size was shown to considerably affect the shoot recovery rate from cryopreserved shoot tips. In pear, various sizes of shoot tips were used, e.g., 0.8–1.0 mm [19], 1.5–2 mm [36], and 2.5–3 mm [37]. However, no comparative studies on the effect of shoot size on the shoot recovery rate had been conducted. In the present study, we found that the optimal shoot tip size for shoot recovery was approx. 1 to 1.5 mm; that means the apical shoot tip with the apical meristem, one or two primordia and the tissue base (Figure 3).

Cold acclimation is very effective for improving the recovery rate of shoot tips from cryopreservation, especially in temperate plants [38]. Strongly varying applications in terms of both temperature and duration were used in *Pyrus;* 8–12 weeks of cold acclimation at 0 °C of donor cultures [37], 3 weeks at 5 °C [36], alternating temperature (16 h at −1 °C darkness, and 8 h at 22 °C light) for 2 to 5 weeks [19]. In the present experiments (Table 4), a two-week cold pre-treatment with alternating temperatures of the in vitro cultures was used, analogously to the last reference. Based on these results, there are further possibilities for future experiments, especially as very long cold pre-treatments of up to 15 weeks prove to be positive, depending on the species [19]. Further optimization of recovery was reached for *P. cordata* accession by using 3 weeks of culture on 50 µM abscisc acid or 5–7% sucrose medium followed by 2 weeks of low alternating temperature [39].

In the vitrification-based methods for shoot tip cryopreservation, the exposure time to the PVS2 or comparable substances is a critical point for the recovery rate after cryopreservation [40]. For dehydration in *Pyrus,* PVS2 was used for 80 min at 25 °C [36] or for an immersion time of up to 15 min refrigerated by a crushed-ice bed [41]. We applied PVS2 solution exposure on ice as it could enable a slow penetration of cryoprotectants into the tissues to alleviate osmotic stress and toxicity. According to the results of Table 3, it can be seen that depending on the genotype, the control without cryopreservation also shows reduced values of regeneration. For this reason, other vitrification solutions must be considered for future experiments, or the times of exposure must be varied.

In the case of the *Pyrus* wild species collection, two alternative techniques for long-term storage using cryopreservation were tested for the first time under the condition of the ‘Fruit Genebank’ Dresden-Pillnitz, i.e., PVS2 vitrification of shoot tips and dormant buds. A total of 35 accessions of 21 *Pyrus* species were included in all experiments. Even after these initial experiments, this was seen to offer not only a unique opportunity to compare the efficiency in terms of recovery results, but also in terms of time and labour requirements. According to the comparison in Table 4, where the results of *Pyrus* using the same genotypes for both methods of cryopreservation are summarized, no clear conclusion regarding the more effective method can be drawn. With these initial results, it could be demonstrated that with the PVS2 method a higher number of genotypes reached the required baseline of 40%. Further experiments using both techniques are planned for the future.

Comparative studies with ancient apple cultivars of Vento, Italy showed higher regeneration rates with the PVS2 method [42]. Even, post-harvest cold-exposure of scion wood was used as in our experiments to increase cold hardiness of dormant buds. Chen et al. came to the same conclusion when using the apple cultivar ‘Golden Delicious’ [43]. On the other hand, the advantages of the dormant bud method are well known, the use of dormant buds costs less, is simpler to accomplish, and requires less time than using in vitro shoots. No in vitro culture is included and a further benefit is the short time needed to regenerate a grafted tree [27]. The weaknesses of the method are its dependence on the year and the limited possibility of modifying the process methodically.

Until now, no studies have been carried out by us on genetic stability after cryopreservation. However, literature data show that no morphological and genetic deviations could be detected in either *Malus* [44] or *Pyrus* [45].

## 5. Conclusions

A unique and valuable diversity of the *Malus* and *Pyrus* wild species germplasm is maintained ex situ in field collections in the ‘Fruit Genebank’ in Dresden-Pillnitz, Germany, with two trees per accession. Cryopreservation can be the method of choice in this case for duplication. However, a wider application of plant cryopreservation depends on the availability of efficient and reproducible protocols applicable to many different plant species of the genera *Malus* and *Pyrus*.

Based on the studies presented, a total of 180 *Malus* accessions belonging to 32 species were processed with an average recovery rate of 39% within ten years. Accessions, 116 in number, achieved the criterion of 40% recovery which corresponds to 64.44% of all accessions tested. Despite the fact that dormant buds present an experimental system with uninfluenceable sources of variation, it is a method applicable to a wide range of *Malus* genotypes and will be used annually to test other species of the genus and thus duplicate the *Malus* collection in the cryo-collection.

In the case of the *Pyrus* wild species collection, two alternative techniques for long-term storage using cryopreservation were tested for the first time under the conditions of the Dresden-Pillnitz ‘Fruit Genebank’, namely PVS2 vitrification of shoot tips and dormant buds. A total of 35 accessions of 21 species was included in all experiments in *Pyrus.* Although the first comparative experiments demonstrated that the PVS2 method resulted in a larger number of genotypes which reached the required baseline of 40%, further experiments using both techniques are planned for the future to make a final conclusion on the method of choice.

## Figures and Tables

**Figure 1 biology-12-00200-f001:**
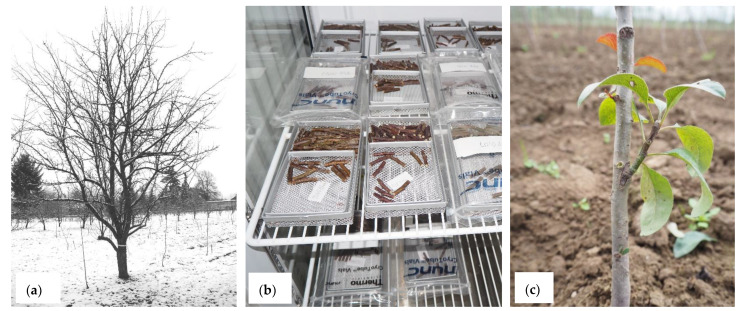
Cryopreservation of dormant *Pyrus* buds: (**a**) Trees of *P. pyraster* during winter-time in January; (**b**) Single bud sections during the dehydration step at a temperature of –5 °C; (**c**) Recovery of *P. elaeagrifolia* buds after cryopreservation in the orchard.

**Figure 2 biology-12-00200-f002:**
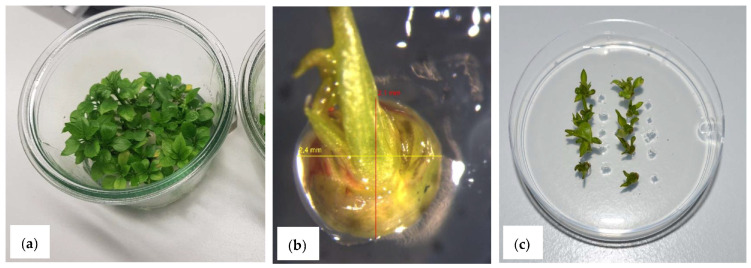
Cryopreservation of *Pyrus* by using PVS2 vitrification: (**a**) Donor material in vitro shoots; (**b**) Explant of *P. korshinskyi-*shoot tip with four primordia; (**c**) Recovery of *P.* ×*bretschneideri* shoot tips after cryopreservation.

**Figure 3 biology-12-00200-f003:**
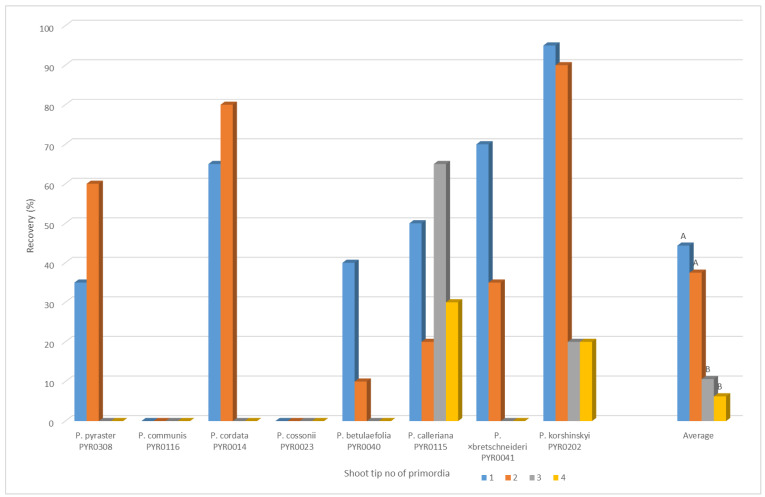
Cryopreservation of *Pyrus* by using PVS2 vitrification: Recovery rate (%) of *Pyrus ssp*. shoot tips after cryopreservation depending on the size of the explant. For each variant = 20 shoot tips. Means with the same letters are not significantly different (*p* ≤ 0.05).

**Table 1 biology-12-00200-t001:** Mean recovery rates after cryopreservation (%) of dormant bud explants is dependent on the *Malus* species. Rootstock for chip budding was M9. The taxonomic classification is according to Forsline et al. [23].

Section	Species	No Tested Acc.	No Acc. Recovered	No Acc. Recovery ≥ 40%	% of Acc. Recovery ≥ 40%	Mean of Recovery %	SD. of Recovery
Malus	*M. pumila*	2	2	2	100.00	65.00	20.00
	*M. prunifolia*	49	48	41	83.67	63.00	25.73
	*M. ×domestica*	34	34	25	73.53	57.56	28.86
	*M. orientalis*	3	3	2	66.70	46.50	29.80
	*M. sieversii*	30	28	15	50.00	43.58	28.71
	*M. sylvestris*	4	4	2	50.00	42.50	28.61
	*M. ×moerlandsii*	3	2	0	0.00	24.07	17.17
	*M. spectabilis*	3	3	0	0.00	5.00	0.00

Baccatus	*M. baccata*	4	3	3	75.00	72.50	42.06
	*M. ×robusta*	3	3	2	66.70	65.47	22.04
	*M. hupehensis*	2	2	1	50.00	38.60	33.60
	*M. sikkimensis*	1	1	0	0.00	22.20	
	*M. ×hartwigii*	1	0	0	0.00	0.00	0.00
	*M. halliana*	1	0	0	0.00	0.00	0.00

Sorbomalus	*M. ×sublobata*	3	3	2	66.70	73.33	27.18
	*M. komarovii*	3	3	2	66.70	58.33	24.61
	*M. ×zumi*	3	3	2	66.70	55.00	24.83
	*M. prattii*	1	1	1	100.00	50.00	
	*M. sieboldii*	3	3	2	66.70	47.97	24.01
	*M. floribunda*	4	4	2	50.00	36.50	31.05
	*M. transitoria*	2	2	2	50.00	30.00	25.00
	*M. toringoides*	3	3	1	33.30	22.60	12.51
	*M. sargenti*	3	3	3	0.00	16.50	15.84
	*M. fusca*	3	2	0	0.00	15.73	11.18
	*M. florentina*	2	2	0	0.00	14.05	8.15
	*M. yunnanensis*	1	1	0	0.00	10.00	

Chloromeles	*M. ×soulardii*	1	1	1	100.00	95.00	
	*M. ×dawsoniana*	1	1	1	100.00	88.90	
	*M. coronaria*	2	2	2	100.00	52.50	12.50
	*M. ioensis*	2	2	2	100.00	50.00	10.00

Docyniopsis	*M. tschonoskii*	1	0	0	0.00	0.00	0.00
Eriolobus	*M. trilobata*	2	0	0	0.00	0.00	0.00

Total		180	169	116	64.44	39.00	18.65

**Table 2 biology-12-00200-t002:** Mean recovery rates after cryopreservation (%) of dormant bud explants is dependent on the *Pyru*s species. Seedling rootstock for chip budding was ‘Kirchensaller Mostbirne’. The taxonomic classification is according to Phipps et al. [24].

Section	Subsection	Species	No Tested Acc.	No Acc. Recovered	No Acc. Recovery ≥ 40%	Mean of Recovery %	SD. Recovery
Pyrus		*Pyrus balansae*	1	1	1	65.00	
		*Pyrus caucasica*	2	2	2	42.50	2.50
		*P. pyraster*	9	9	4	35.00	15.46
		*P. communis*	1	1	0	5.00	

Pashia Koehne	Armoricana (Decne.) Terpó	*P. cossonii*	1	1	1	75.00	
		*P. cordata*	1	1	1	65.00	

	Mongolica (Decne.) Terpó	*P. aromatica*	1	1	1	45.00	
		*P. asia-mediae*	1	1	0	35.00	
		*P. lindleyi*	1	1	0	5.00	
		*P. ovoidea*	1	1	0	5.00	

	Pashia (Koehne) Terpó	*P. betulaefolia*	4	2	1	11.25	16.72
		*P. ×phaeocarpa*	1	1	0	5.00	
		*P. pashia*	1	0	0	0.00	

	Pontica Decne. Emend. Terpó	*P. elaeagrifolia*	2	2	0	22.50	12.50
		*P. salicifolia*	1	0	0	0.00	

	Pyrifolia Tuz	*P. ×bretschneideri*	2	2	1	17.50	12.50

	Xeropyrenia (Fed.) Tuz	*P. heterophylla*	1	1	0	5.00	
		*P. korshinskyi*	1	1	0	5.00	
		*P. regelii*	1	0	0	0.00	
Total			33	28	12	25.45	

**Table 3 biology-12-00200-t003:** Mean recovery rates after cryopreservation (%) by using PVS2 vitrification of shoot tips is dependent on the *Pyru*s species. In the first two years, controls without cryopreservation were additionally analyzed for some accessions.

Section	Subsection	Species	No Acc.	No of Ex-Plants	Recovery (%)		
					Cryo	SD	Control without Cryo
Pyrus		*P. pyraster*	PYR0312	20	70.00		
		*P. pyraster*	PYR0303	20	60.00		
		*P. pyraster*	PYR0308	80	43.75	20.12	
		*P. pyraster*	PYR0308	10			40.00
		*P. communis*	PYR0116	80	11.25	14.31	
		*P. communis*	PYR0116	20			15.00
		*P. balansae*	PYR0020	40	0.00		

Pashia Koehne	Armoricana (Decne.) Terpó	*P. cordata*	PYR0014	80	48.75	19.49	
		*P. cordata*	PYR0014	10			70.00
		*P. cossonii*	PYR0023	100	0.20	0.40	
		*P. cossonii*	PYR0023	10			10.00

	Mongolica (Decne.) Terpó	*P. ovoidea*	PYR0032	20	5.00		

	Pashia (Koehne) Terpó	*P. betulaefolia*	PYR0040	80	28.75	14.31	
		*P. betulaefolia*	PYR0040	20			40.00
		*P. calleriana*	PYR0115	80	30.00	23.72	
		*P. calleriana*	PYR0115	10			30.00

	Pontica Decne. Emend. Terpó	*P. ×nivalis*	PYR0073	20	10.00		
		*P. elaeagrifolia*	PYR0017	20	15.00		

	Pyrifolia Tuz	*P. ×bretschneideri*	PYR0041	100	47.00	16.31	
		*P. ×bretschneideri*	PYR0041	20			95.00

	Xeropyrenia (Fed.) Tuz	*P. korshinskyi*	PYR0202	80	88.75	8.20	
		*P. korshinskyi*	PYR0202	10			30.00
Average					32.67		40.78

**Table 4 biology-12-00200-t004:** Comparison between PVS2 vitrification of shoot tips and dormant bud techniques depending on *Pyru*s species. Shown are the regeneration values for the maximum and the mean value for PVS2 as well as the regeneration value for the dormant buds *.

Section	Subsection	Species	No Acc.	Recovery (%)		
				PVS2 Maximum	PVS2 Average	Dormant Buds
Pyrus		*P. pyraster*	PYR0312	70.00		15.00
		*P. pyraster*	PYR0308	65.00	43.75	50.00
		*P. pyraster*	PYR0303	60.00		30.00
		*P.* *communis*	PYR0116	35.00	11.25	5.00
		*P. balansae*	PYR0020	0.00	0.00	65.00

Pashia Koehne	Armoricana (Decne.) Terpó	*P. cordata*	PYR0014	75.00	48.75	65.00
		*P. cossonii*	PYR0023	5.00	0.20	75.00

	Mongolica (Decne.) Terpó	*P. ovoidea*	PYR0032	5.00		5

	Pashia (Koehne) Terpó	*P. betulaefolia*	PYR0040	45.00	28.75	0

	Pontica Decne. Emend. Terpó	*P. elaeagrifolia*	PYR0017	15.00		10.00

	Pyrifolia Tuz	*P. ×bretschneideri*	PYR0041	70.00	47.00	5.00

	Xeropyrenia (Fed.) Tuz	*P. korshinskyi*	PYR0202	95.00	88.75	5.00
Average				45.00	33.56	27.50

* The number of shoot tips or dormant buds corresponds to the data in Table 2 and Table 3. The best variant was marked in grey.

## Data Availability

Not applicable.

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
