# Peer review of "Cryopreservation of Malus and Pyrus Wild Species in the ‘Fruit Genebank’ in Dresden-Pillnitz, Germany"

_biology, 2023, doi:10.3390/biology12020200_

Round 1
Reviewer 1 Report
Dear Authors,
The manuscript entitled “Cryopreservation of genetic resources for Malus and Pyrus wild species in the ‘Fruit Genebank’ in Dresden-Pillnitz, Germany” has been reviewed in detail. In the study, only samples belonging to the genus Pyrus were cryopreserved using two different methods. On the other hand, samples of Malus genus were cryopreserved only via dormant bud cryopreservation technique. This gives the impression that the study is not yet completed. The manuskript may be published in the journal after some of the corrections and suggestions I have listed in the report.
Kind regards

Author Response
Please, see the attached file.

Reviewer 2 Report
The present ms desribes valuable work performed with wide and unique genetic materials. The introduction is well written and gives a good background for the study, with clearly indicated goals.
The major shortcomings with the present ms are related to presentation of material and methods, mainly to control treatments done (based on results) but not described in M&M. In more detail, please add the following:
- when describing Pyrus materials for PVS2 experiments (r139 onwards), give reference to used micropropagation method or describe it shortly, including initiation and proliferation of cultures. Now only media and culture room conditions are mentioned
- likewise, when desribing the PVS2 regeneration (r183 onwards) describe or give reference to proliferation medium. Was the transfer to light performed at once or gradually ?
- add information on PVS2 control treatment (r199) !! Was it PVS2 treatment without LN or what ? Give details.
- add information on controls done for Malus dormant bud experiments! These are suddenly mentioned at results (r221) but not explained in M & M i.e. add info on these. Was the control only for moisture content, or did you have ontrols for chip budding with and without cryo using Idared cultivar too ?? If no control were performed, could you anyhow tell the overall success rates for chip budding using non-cryostored material e.g. in discussion?
- further, the results of explant size experiment appear suddenly at r252. This experiment needs to be explained also in M & M at an appropriate point when describing PVS2 experiments
- it would be good to indicate the lenght of cryostorage for the materials e.g. in tables, as it apparently varies from accession to accession. The phrases like "long-term cryopreserved shoot tips "(r265) would have more meaning. I hope that you have looked at if there is a connection between storage time and recovery ? This could be reported too.
- minor issue: "scion wood / bud wood" is not the best term to describe twigs or branches with dormant buds used for experiments. Consider changing it.
At discussion, the studied alternative options for cryostorage are compared. How about cryopreservation of dormant buds, and then their regeneration using micropropagation instead of chip budding, i.e. initiation of cultures from cryostored and thawed buds. Would be interesting to have a short comment on potentials and shortcomings of this approach too.
Author Response
Please, see the attached file.

Reviewer 3 Report
The introduction is logically presented, but is a little bit redundant, because it includes basic biologic information that is well known by most of the potential readers. I would suggest to reduce it to the information that is not common background to all practitioners.
Some of the background information presented in the introduction is repeated also in the Conclusions section. It should be removed from there.
The experiments reported in this paper do not address the possible induction of genetic variation after cryopreservation (see for instance Yunguo Liu, Xiaoyun Wang, Lingxiao Liu, Analysis of genetic variation in surviving apple shoots following cryopreservation by vitrification, Plant Science, Volume 166, Issue 3, 2004, Pages 677-685 or Adela Halmagyi, Constantin Deliu, Valentina Isac, Cryopreservation of Malus cultivars: Comparison of two droplet protocols, Scientia Horticulturae, Volume 124, Issue 3, 2010, Pages 387-392). Experiments of long-term cryopreservation should include tests to exclude genetic variation in the regenerated plants. This aspect should be at least discussed in the conclusion section.
PSV2 should be spelt out the first time that is used (Preparing Vitrification Solution).
Author Response
Please, see the attached file.

Reviewer 4 Report
the article titled " Cryopreservation of genetic resources for Malus and Pyrus wild species in the ‘Fruit Genebank’ in Dresden-Pillnitz, Germany". I read the manuscript and It looks good contains a very important contents. But I have some general comments such as:
Introduction is too long and should be summarized.
Conclusion is very long and should be summarized.
Materials and methods should contain;
1- The identification of the two wild types using molecular biology using ITS, 18S etc.
2- Karyotypes differentiation for the two wild types compared with cultivated ones.
3- Somatic variation occurred during the dormant buds of Malus and Pyrus preservation method.
4- Complete analysis for the fruit components in the plant resulted from the two methods of preservation.
5- The optimum conditions and the lonetivity of each method used in the plant materials preservation.
Author Response
Please, see the attached file.

Round 2
Reviewer 1 Report
Dear Author,
Your manuscript can be published in its current form.
Kind regards
Author Response
The manuscript can be published in its current form.
Reviewer 2 Report
I am happy to see that the authors have taken my comments into account, at least partly. They still need to revise or clarify the following points:
Orig. comment: likewise, when desribing the PVS2 regeneration describe or give reference to proliferation medium.
à the authors now refer to “the same” proliferation medium (r178). This is a bit difficult to interpret since the only medium mentioned in the chapter 2.3. is the MS medium (Duchefa) supplemented with 5 % (v/v) dimethyl sulfoxide (DMSO; neoLab, Heidelberg, Germany) and 1 g of additional agar(0.85 % [w/v]; DifcoTM Agar granulated, BD) for 2 d under cold-acclimation conditions. Surely not medium with DMSO and additional agar was used for regeneration ? Please clarify !
Orig. comment: - add information on controls done for Malus dormant bud experiments! These are suddenly mentioned at results but not explained in M & M i.e. add info on these. Was the control only for moisture content, or did you have ontrols for chip budding with and without cryo using Idared cultivar too ?? If no control were performed, could you anyhow tell the overall success rates for chip budding using non-cryostored material e.g. in discussion?
à based on revised M&M (r 125-126) the Ida red variety controls were done for the whole process. So, please add also the control results to the results section. Now only moisture content is given for IdaRed material, but it is important to give the recovery rates for it too.
Orig. comment: - further, the results of explant size experiment appear suddenly at r252. This experiment needs to be explained also in M & M at an appropriate point when describing PVS2 experiments
à this is now mentioned in the M & M (r.194-195) but not explained properly. I.e. describe how buds were classified into different size classes, how many replicates per genotypes etc.
Orig. comment: - it would be good to indicate the lenght of cryostorage for the materials e.g. in tables, as it apparently varies from accession to accession. The phrases like "long-term cryopreserved shoot tips "(r265) would have more meaning. I hope that you have looked at if there is a connection between storage time and recovery ? This could be reported too.
When stored in liquid nitrogen, all biophysical and biochemical processes are stopped, so storage is assumed to be unlimited. Studies by Caswell and Kartha (2009) on strawberries have shown that after 28 years of storage in liquid nitrogen, no changes in regeneration occur. So far, I am not informed of any long-term studies on apples and pears.
à I was not asking for adding literature on the topic, but simply to mention how long time your material was cryostored. I see that at the minimum of 1 day (as now stated in r173) but to show the variation that you had, as apparently all the samples were not stored for 1 day ?? And, if you had a lot of variation in cryostorage time (from one day to weeks or months or years even ?), please tell if you have checked the potential effect of storage time on the recovery ? There are recent papers reporting that recovery rates in woody plants may decrease with increasing storage time including discussion on potential reasons for this.. see eg. Varis et al. 2022
Orig. comment: At discussion, the studied alternative options for cryostorage are compared. How about cryopreservation of dormant buds, and then their regeneration using micropropagation instead of chip budding, i.e. initiation of cultures from cryostored and thawed buds. Would be interesting to have a short comment on potentials and shortcomings of this approach too.
The comment has been taken into account and processed.
è Good that this is now mentioned as an option in the discussion – however, without any comment on its potential or shortcomings compared with chip-budding technique that I was looking for. I.e it would be good to see some discussion if regeration of cryostored dormant buds either via chip-budding or micropropagation could have any advantages or shortcomings
Author Response
Please see the enclosed document!

Reviewer 3 Report
The sentences added by the authors to address my comment are contradictory:Until now, no studies have been carried out on genetic stability after cryopreservation. However, literature data show that no morphological and genetic deviations could be detected in either Malus [44] or Pyrus [45].
The authors should specify that studies on genetic stability after cryopreservation have not been carried out by themselves. Otherwise the first part of the paragraph is the contrary of the second sentence.
Author Response
I changed the recommended sentences. 'Until now, no studies have been carried out by ourselves on genetic stability after cryopreservation.'